# Potential Therapies Using Myogenic Stem Cells Combined with Bio-Engineering Approaches for Treatment of Muscular Dystrophies

**DOI:** 10.3390/cells8091066

**Published:** 2019-09-11

**Authors:** Norio Motohashi, Yuko Shimizu-Motohashi, Thomas C. Roberts, Yoshitsugu Aoki

**Affiliations:** 1Department of Geriatric Medicine, Tokyo Metropolitan Institute of Gerontology, Tokyo 173-0015, Japan; 2Department of Molecular Therapy, National Institute of Neuroscience, National Center of Neurology and Psychiatry, Tokyo 187-8502, Japan; 3Department of Child Neurology, National Center Hospital for Neurology and Psychiatry, National Center of Neurology and Psychiatry, Tokyo 187-8502, Japan; 4Department of Paediatrics, University of Oxford, South Parks Road, Oxford OX1 3QX, UK

**Keywords:** skeletal muscle, stem cells, regeneration, dystrophy, therapy

## Abstract

Muscular dystrophies (MDs) are a group of heterogeneous genetic disorders caused by mutations in the genes encoding the structural components of myofibres. The current state-of-the-art treatment is oligonucleotide-based gene therapy that restores disease-related protein. However, this therapeutic approach has limited efficacy and is unlikely to be curative. While the number of studies focused on cell transplantation therapy has increased in the recent years, this approach remains challenging due to multiple issues related to the efficacy of engrafted cells, source of myogenic cells, and systemic injections. Technical innovation has contributed to overcoming cell source challenges, and in recent studies, a combination of muscle resident stem cells and gene editing has shown promise as a novel approach. Furthermore, improvement of the muscular environment both in cultured donor cells and in recipient MD muscles may potentially facilitate cell engraftment. Artificial skeletal muscle generated by myogenic cells and muscle resident cells is an alternate approach that may enable the replacement of damaged tissues. Here, we review the current status of myogenic stem cell transplantation therapy, describe recent advances, and discuss the remaining obstacles that exist in the search for a cure for MD patients.

## 1. Introduction

Muscular dystrophies (MDs) consist of several genetic diseases that are caused by a variety of mutations in structural proteins, and that result in progressive degeneration and weakness due to muscle damage, inflammation, or deposition of adipose and fibrotic tissue [1]. Several types of MDs exist, including Duchenne muscular dystrophy (DMD), Becker muscular dystrophy (BMD), Facio-scapulohumeral muscular dystrophy (FSHD), myotonic dystrophy, limb-girdle muscular dystrophies (LGMDs), Emery-Dreifuss muscular dystrophy (EDMD), and Fukuyama congenital myopathy (FCMD).

DMD is an X-linked skeletal myopathy that affects approximately 1 in 5000 male births [2] and manifests as progressive muscle weakness and atrophy, ultimately resulting in failure of respiratory and cardiac functions [3]. DMD is caused by a mutation in the *DMD* gene, which leads to a deficiency of functional dystrophin protein at the plasma membrane [4,5]. Dystrophin is a critical component of the dystrophin-glycoprotein complex (DGC) that links the cytoskeleton and extracellular membrane. DGC insufficiency leads to muscle fragility and contraction-induced damage [6]. BMD is also caused by mutations in the *DMD* gene, but myofibrils retain an internally truncated form of the dystrophin protein, resulting in milder symptoms with late disease onset and relatively advanced survival age [7]. FSHD is a severe form of muscular dystrophy characterized by asymmetric and progressive atrophy and weakness of skeletal muscles of the face, scapula, and upper arms [8]. LGMD was designated a separate entity from X-linked dystrophinopathies such as DMD and BMD by Walton and Nattrass in 1954 [9]. LGMD progresses slowly, but eventually leads to severe disablement and often premature death. Autosomal EDMD is caused by mutations in the Lamin A/C (LMNA) gene and is characterized by progressive wasting and weakness in scapulo-humero-peroneal muscles [10,11]. Fukuyama type muscular dystrophy (FCMD) is a congenital progressive muscular dystrophy characterized by motor impairment, dystrophic changes in skeletal muscle, severe intellectual deficit, and brain malformation [12].

Currently, there are no definitive cures for MDs. So far, steroids are the only standardized therapy for DMD and are used to delay disease progression [13]. However, steroids are associated with a risk of severe side effects, including bone and epidermal thinning, hypertension, mood/behaviour changes, dysregulated metabolism, delayed puberty, and stomach irritation, among others. At present, MD therapies, including exon skipping, stop codon read-through, and viral vector-based approaches, are most advanced for DMD, and some have progressed to the clinical trial stage [14,15]. However, these treatments have limited efficacy as well as the potential to elicit adverse immune responses and are unlikely to be curative. Stem cell–based therapy is a promising approach that has the potential for broad application in the treatment of MDs. In this review, we describe emerging MD therapies, with a particular focus on stem cell-based therapies, and future treatment prospects.

## 2. Current Status of Myogenic Cell Therapy

Skeletal muscle has the remarkable potential to regenerate and can recover rapidly following muscle injury. Muscle resident stem cells, particularly satellite cells [16], play a central role in muscle regeneration; therefore muscle satellite cells and cultured satellite cell (myoblast)-based therapies are considered to be a promising approach for treating MDs [17]. Since the early 1990s, cell transplantation has been attempted for DMD to restore functional dystrophin protein. Myoblasts obtained by culturing muscle satellite cells from wild-type mice implanted into skeletal muscle of *mdx*, a DMD model mouse, have enabled the successful reconstruction of dystrophin protein [18,19]. These studies were followed by a phase I clinical trial, where allogeneic myoblasts were transplanted into DMD patients, and as expected, dystrophin protein expression was derived from transplanted cells [20,21,22,23]. Further clinical trials for myoblast transplantation are ongoing and are expected to be completed in 2019 (NCT02196467). However, the cell transplantation efficiency in these studies is quite low, and in order to improve sufficient dystrophin expression and muscle strength, further work is required.

A possible reason for this insufficient expression could be a low survival rate of implanted cells due to immuno-rejection of healthy donor cells. This problem was partially solved through the use of immunosuppressants, including calcineurin inhibitors, and tacrolimus in particular [24,25,26,27]. Experiments using tacrolimus in mice and monkeys as well as clinical trials using immunosuppressants resulted in a significant improvement in transplantation efficiency [24,25,26,27]. However, the risk of infection due to long-term administration of immunosuppressants remains a concern [28].

In order to avoid possible immune responses, autologous transplantation of genetically modified myoblasts was performed [29,30,31]. Engraftment of myoblasts harbouring a genetically altered form of the micro-dystrophin gene successfully restored dystrophin protein in mice and in monkeys [31]. However, transplantation of autologous myoblasts with genetic manipulations may be potentially limited due to gradual senescence of myogenic cells in ageing patients. Interestingly, a decline in replicative capacity was observed with increasing donor age, and was accelerated in DMD myoblasts as compared to control myoblasts [32]. Freshly isolated satellite cells possessed a high capacity to re-constitute muscle fibres and quiescent satellite cells after engraftment, but this capacity declined following expansion and passage of myoblasts in vitro [33]. Therefore, both un-modified myoblast cells from healthy donors with a normal *DMD* gene and genetically modified autologous transplantation of cells present drawbacks, either from risk of immune rejection or from required manipulation of the gene in advance of engraftment, respectively [34].

Other possible reasons for transplantation insufficiency could be related to the following: (1) low survival rate of implanted cells due to apoptosis or necrosis, (2) low potencies of myoblast proliferation, differentiation, and migration, and (3) low capacity of self-renewal to replenish the satellite cell pool. In addition, satellite cells or myoblasts are unable to cross the endothelium, and the systemic delivery via blood vessels is limited, making intravascular transplantation of these cells technically challenging [35]. 

Given the difficulty of intra-vascular transplantation, clinical application of this technique would be challenging for several types of MDs, including DMD, since the majority of muscles in MDs are affected. However, intramuscular injection of myogenic cells is required, since MDs with locally affected muscles, such as oculopharyngeal muscular dystrophy (OPMD), may benefit from a local injection of myogenic cells. Autologous myoblast transplantation in OPMD patients via local intramuscular injections was conducted as a phase I/IIa clinical trial (NCT00773227) [36]. Muscle satellite cells derived from non-affected muscle in the same patients were expanded in vitro without genetic corrections and were injected into severely damaged pharyngeal muscles of patients [36]. Although a massive amount of donor cells was required for cell survival in dystrophic muscle, local administration of myoblasts led to improvement of muscle regeneration and had a beneficial effect on swallowing function [36]. Additionally, a study of intra-arterial injection of HLA-matched human mesoangioblasts into DMD patients proved to be feasible and relatively safe without adverse events such as inflammation or tumour formation [37]. Thus, the delivery strategy and specific cells used for therapy should be modified depending on the type of MD.

## 3. Cell Sources for Transplantation Therapy

The most significant challenges for cell therapy are related to the limited efficiency of transplantation and systemic delivery of myogenic cells. These challenges have led researchers to seek novel candidate cell sources that have the potential to differentiate into myogenic cells. In addition to muscle satellite cells, other myogenic progenitor cells have been identified within skeletal muscle, including side population (SP) cells [38,39], muscle-derived stem cells (MDSC) [40], multipotent adult precursor cells (MAPC) [41], myogenic-endothelial progenitors [42], CD133+ stem cells [43,44,45,46], mesoangioblasts [37,47,48,49], or pericytes [50] that are multipotent and have capacity to differentiate into several cell types (Figure 1). These cells have the potential to differentiate into skeletal muscle both in vitro and in vivo. A notable characteristic of SP cells, CD133+ stem cells, mesoangioblasts, and pericytes is that, unlike satellite cells or myoblasts, they can pass through vascular walls into muscles, thus enabling systemic delivery. 

CD133+ cells are blood- [43] and muscle-derived [44] stem cells that are able to differentiate into multiple cell types, including muscle, hematopoietic, and endothelial cells. CD133+ cells originating from the blood and muscle of mice and humans can migrate through blood vessel walls, and have been shown to contribute to muscle fibre regeneration following implantation into *mdx* mice [43,45]. A clinical trial has demonstrated that autologous intramuscular transplantation of muscle-derived CD133+ stem cells could form myofibres and increase capillary number without any local adverse effects, indicating that intramuscular injection of autologous stem cells is safe and effective [44]. Importantly, intra-arterial transplantation of genetically modified blood- or muscle-derived CD133+ cells from DMD patients enabled muscle regeneration in *mdx* mice [45]. Additionally, human muscle-derived CD133+ cells re-constituted satellite cells after intramuscular transplantation in mice [46]. However, the myogenic capacity of CD133+ cells from DMD patient muscles was lower than those from normal human muscles, and contributed significantly less to muscle regeneration [51]. These results should be taken into account when developing stem cell therapies.

Mesoangioblasts, which were identified in the wall of mouse embryonic dorsal aorta, were found to participate in the postembryonic development of the mesoderm [48]. Additionally, a previous study found that mesoangioblasts could differentiate into muscle fibres in vitro and in vivo [47]. Normal canine mesoangioblasts delivered via arterial injection generated dystrophin-expressing fibres in dystrophic model dogs, resulting in the amelioration of disease-related changes to muscle morphology [49]. In a clinical trial, human mesoangioblasts derived from human leukocytes taken from an antigen-matched healthy brother were transplanted arterially into patients with DMD. This trial demonstrated the relative safety of human mesoangioblasts without any adverse events, although clinical efficacy, as measured by myofibre-forming ability of injected cells, was not apparent [37].

Pericytes are located beneath the basal lamina of small vessels in adult tissues and are thought to derive from mesoangioblasts developmentally [50]. Although these cells did not express myogenic markers (Pax7, Myf5 and MyoD), they could differentiate into dystrophin-expressing muscle fibres after intra-arterial injection into immunodeficient-*mdx* (SCID-*mdx*) mice [50]. In addition, human pericytes from DMD patients, in which human mini-dystrophin gene was transduced by a lentiviral vector, caused the generation of myofibres expressing human dystrophin when injected into SCID-*mdx* mice intra-muscularly [52].

As described above, both CD133+ cells and mesoangioblasts can differentiate into myogenic cells and are able to penetrate blood vessel walls. Additionally, mesoangioblasts express immunologically relevant molecules. In fact, infiltration of dystrophin-reactive T cells was observed in DMD muscles, and a robust T cell response may affect the success of therapeutic approaches [53]. Myoblasts have the potential to trigger a T-cell mediated immune responses during regeneration of muscle tissue due to an increase of MHC (major histocompatibility complex) I/II expression induced by inflammatory factors, including IFNγ in chronic inflammatory muscle disorders [54,55,56]. On the other hand, mesoangioblasts and iPS cell-derived mesoangioblast-like cells have been shown to induce the expression of PD-1/PD-L1, an immune response suppressor, and confer resistance to T-cell killing through the production of proinflammatory cytokines [57,58]. These observations indicate that immunological factors can affect cell survival following cell transplantation with different myogenic stem cells.

However, the entrapment of delivered cells in filter organs is a possible concern. According to previous studies, greater than 30% of mesoangioblasts were detected in the liver, lungs or spleen of α-sarcoglycan deficient mice 24 hours after arterial injection, and their numbers were even higher 8 months post-injection [59]. Further, about 90% of mesoangioblasts or pericytes from human muscles were localized in filter organs of SCID-*mdx* mice 24 hours after arterial injection [50]. After abdominal aorta injection, trapped human mesenchymal stem cells (MSCs) caused interruption of blood flow through the microvasculature downstream of the site of injection, resulting in pulmonary embolism and subsequent death in up to 40% of SCID mice [60]. These results indicate that arterial delivery of myogenic stem cells may also disrupt arterial or microvascular blood flow, leading to tissue damage.

So far, CD133+ cells and mesoangioblasts derived from healthy donors have been used in preclinical studies [37,44]. Adverse events, including inflammation, tumour formation or tissue infarction, have not been reported after intra-arterial injection of human mesoangioblasts [37] and intramuscular injection of human muscle CD133+ cells [44]. Therefore, although results from animal and clinical experiments indicated a low engraftment efficacy, these cells are still considered promising candidates to treat patients with DMD.

The generation of sufficient numbers of transplantable cells remains a significant hurdle to be overcome for cell therapy. Embryonic stem (ES) cells or induced pluripotent stem (iPS) cell-based therapies have many advantages, including increased replicative potential. Therefore, these cells were anticipated to be an alternative cell source for DMD therapy (Figure 1). Substantial progress has been made in the development of iPS cell technology, raising the possibility of personalized therapies while eliminating ethical concerns, since patient-specific iPS cells can be easily generated and expanded [61,62,63]. Additionally, patient-derived iPS cells can be genetically corrected with technologies such as adeno-associated virus, lentivirus, TALENs, or the CRISPR-Cas9 system, prior to differentiation into transplantable myogenic progenitor cells (Figure 1). The low risk of immune rejection and the potential for generating an unlimited number of cells indicate that iPS cells and iPS-derived myogenic cells could provide attractive approaches for the treatment of DMD when combined with gene therapy. 

For example, human limb-girdle muscular dystrophy type 2D-iPS cell-derived mesoangioblasts expanded after genetic correction in vitro, were intra-arterially engrafted into α sarcoglycan-deficient mice, and resulted in the reconstitution of sarcoglycan-positive muscle fibres by iPS-derived cells [64]. Similar results were also obtained in another study, in which dystrophic mice-derived iPS cells transduced with a micro-utrophin gene, the homologue of dystrophin, were differentiated into PDGFαR^+^ Flk1^−^ myogenic progenitor cells, and intravenously injected into dystrophic mice [65]. Additionally, a recent study has shown that human DMD iPS cells-derived skeletal muscle cells for which the DMD gene was reframed with the CRISPR-Cas9 system, expressed stable dystrophin that improved membrane stability and restored the DGC member β-dystroglycan after intramuscular injection [66]. These reports provide proof-of-concept for the potential utility of systemically delivered iPS-derived cells in DMD mouse models. Nonetheless, even with the advantages of using iPS cells, cell transplantation efficiency in these reports was still low, and systemic delivery, as well as engraftment efficiency, remained a problem. Moreover, the production of undesired un-differentiated or heterogeneous cell populations containing myogenic cells and other cell types originating from iPS cells may lead to unexpected results, including the risk of tumorigenesis after cell delivery [67]. In order to eliminate possible concerns and to produce clinical-grade cells, more efficient myogenic induction protocols along with the identification of specific markers to purify myogenic cells have recently been developed [68,69,70]. Further investigations are required to identify ideal iPS cell-derived myogenic cell sources capable of successful engraftment with low risk of tumorigenesis following cell delivery. 

## 4. Factors That Affect Cell Potential

Transplantation of myogenic cells into dystrophic muscle requires a sufficient number of cells. Therefore, donor cells are required to be effectively expanded in vitro. However, because proliferation, differentiation, and self-renewal of cells decrease in vitro due to a lack of niche signalling, a method to efficiently expand donor cells while maintaining their myogenic capacity is needed [33,71,72].

According to previous studies, myoblasts that are cultured in tissue culture plates have decreased myogenic differentiation potential due to a loss of myogenic potential and stemness, when compared to freshly isolated muscle satellite cells [33,71]. These results indicate that a niche-like environment is required for the preservation of myogenic cell characteristics in vitro. In addition, myogenic cell function declines with age [73] and as muscles degenerate [74,75]. These effects are most likely the result of repeated expansions, which shorten telomeres and impair self-renewal mechanisms, leading to the exhaustion of muscle satellite cells [74]. Rejuvenation of myogenic cells by gene modification or biochemical molecules before transplantation is a promising therapeutic approach. However, cellular or niche environmental changes that occur in diseases and ageing are thought to accelerate loss of myogenic cell function during transplantation. A previous study using a mouse model of ageing revealed that systemic factors in young mice mediated by heterochronic parabiosis could restore more youthful states to cells and were able to rejuvenate aged stem cells as well as stem cell niches to enhance muscle regeneration [76]. Additionally, deterioration of the muscle environment in aged muscles reduced the efficiency of myoblast engraftment [77]. In DMD model mice, Cabrera et al. demonstrated that andrographolide, an inhibitor of the pro-inflammatory factor, NF-κB, reduces fibrosis and ECM protein expression, resulting in an increase in myoblast transplantation efficacy [78]. Another study by Brunelli et al. demonstrated that treatment with HCT-1026, a non-steroidal anti-inflammatory drug capable of releasing nitric oxide, could reduce inflammation and fibrosis, resulting in an increase in mesoangioblast injection efficacy after intra-arterial delivery in α-sarcoglycan-deficient mice [79]. As has previously been shown, the microenvironment surrounding myogenic cells can alter stem cell proliferation and differentiation, or the promotion of self-renewal following muscle injury [80,81]. Therefore, multiple factors within stem cell niches, including cell-cell interactions, autocrine/paracrine signalling, or cell-to-matrix interactions, potentially affect the properties of myogenic cells (Figure 1). Given this evidence, a focus on improving the environment, both in cultured donor cells and in the recipient’s muscle, may be necessary to achieve successful results for myogenic cell therapy.

### 4.1. Notch Ligands

Heterochronic parabiosis experiments have led to the identification of factors that could systematically restore muscle regeneration via receptor-ligand interactions, such as in Notch signalling [76]. Notch signalling, mediated by the ligand Delta, is critical for various cell functions, including cell-cell interaction and cell fate determination [82] and plays a crucial role in muscle satellite cell maintenance [83,84]. Activation of Notch signalling by delta-like-1 (Dll1) increases the number of proliferating myogenic cells and promotes muscle regeneration after injury [85], while the disruption of Notch signalling in muscle satellite cells induces a differentiation stage that instead leads to depletion of the satellite cell pool and muscle regeneration failure [83,84]. A previous study showed that Notch ligand (Dll1) enhances the efficacy of myoblast transplantation [86], suggesting that Notch ligands are potentially essential factors that sustain myogenic cell self-renewal during in vitro expansion. Another Notch ligand, *Jagged1*, was elevated in golden retriever muscular dystrophy dogs, and overexpression of *Jagged1* was able to rescue the DMD phenotypic features, suggesting that *Jagged1* also acted to improve myogenic cell potential in vivo [87]. These studies suggest that Notch signalling is crucial for the maintenance of skeletal muscle and myogenic cells. 

Alternatively, another study demonstrated that treatment of Notch-ligands, including Dll1, Dll4, and Jagged1, did not enhance the efficacy of myogenic cell engraftment in vitro, even though the induction of Notch signalling promoted the expansion of Pax7(+)MyoD(−) muscle stem-like cells [88]. Thus, it is likely that Notch ligands are required to maintain myogenic cell self-renewal potency. The effects of Notch-ligands for myogenic cell transplantation should be further investigated.

### 4.2. Microvascular Environment

Given that satellite cells are located in close anatomical proximity to capillaries in normal skeletal muscle, interactions between endothelial cells, pericytes, and muscle satellite cells were predicted to be involved in skeletal muscle homeostasis [89,90,91]. Multiple reports have shown an association between endothelial cells, pericytes, and muscle satellite cells [89,90] and revealed that vascular density influences the number of muscle satellite cells [92]. Additionally, autocrine or paracrine signalling of endothelial cells was required for muscle satellite cell proliferation and maintenance [89,90,91]. Abou-Khalil et al. found that secreted Ang1 from vascular smooth muscle cells or fibroblasts located around satellite cells promoted Pax7 expression, and that Ang1/Tie2 signalling through the ERK1/2 pathway was involved in regulating satellite cell self-renewal [90]. Verma et al. demonstrated that muscle satellite cells recruit endothelial cells by secreting VEGFA from myogenic cells to establish a juxtavascular niche [91]. In addition, the microvasculature-derived Notch ligand Dll4 was required for muscle satellite cell self-renewal [91], suggesting that maintenance and proliferation of satellite cells may be influenced by an interaction between endothelial and satellite cells. Additionally, dystrophin was detected in the vascular smooth muscle of wild-type mice [93,94], and in *mdx* mice, decreased vascular density was observed [95,96]. Since an increased vascular density lessened DMD pathology by increasing the number of muscle satellite cells [92], maintenance of the microvascular environment is a factor that may increase the effectiveness of myogenic cell transplantation.

### 4.3. Extracellular Matrix

The extracellular matrix (ECM) is also known as an essential component of skeletal muscle. The constitution of the ECM is a critical factor in muscle regeneration and satellite cell self-renewal. Matrix metalloprotease (MMP)-2, an ECM remodelling factor that is up-regulated in dystrophic muscles [97,98], is known to play a critical role in myogenesis [99], and the overexpression of MMP-2 enhanced myoblast mobility [100]. Further studies have shown that simultaneous transplantation of myogenic cells and SP cells, which are located in the interstitium of skeletal muscle and express a variety of ECM proteins, could promote the efficacy of myogenic transplantation and muscle regeneration, and that these effects are regulated by MMP-2 expression [98]. Gargioli et al. have also demonstrated that pre-treatment with tendon fibroblasts expressing the angiogenic factors PIGF (placenta growth factor) and a metalloproteinase MMP-9, could improve the vascular network and reduce collagen deposition, resulting in an increase in engraftment efficiency after intra-arterial mesoangioblast transplantation in α-sarcoglycan-deficient mice [101]. These observations indicate that the ECM surrounding myogenic cells may be critical for enhancing myogenic cell potential, and the maintenance of skeletal muscle. Alternatively, changes in the levels of ECM components observed during muscle regeneration and diseases likely also influence myogenic cell proliferation, differentiation, or self-renewal. 

The presence of connective tissue or fibrosis caused by excessive deposition of interstitial ECM, seen in muscular dystrophies [8] as well as ageing muscles [102], has led to permanent scarring and organ dysfunction. Since endomysial fibrosis in the muscles of patients with DMD contributes to muscle weakness and is correlated with clinical severity [103], use of anti-fibrotic agents is a possible approach for DMD treatment. In particular, the transforming growth factor-β (TGF-β) pathway plays an important role in fibrotic tissue formation, and is a major target of antifibrotic approaches. For instance, Halofuginone, which is an inhibitor of TGF-β-mediated collagen synthesis, ameliorated muscle pathology and function in addition to reducing fibrosis in *mdx* mice [104], and is currently in a clinical trial [105]. Treatment with Losartan, an inhibitor of TGF-β activity, was also found to increase transplantation efficacy of myoblast and adipose-derived stem cells in *mdx* mice [106,107]. Additionally, reduction of connective tissue growth factors (CTGF/CCN2) induced by TGF-β, or treatment with a neutralizing anti-CTGF monoclinal antibody (FG-3019), decreased fibrosis and increased the efficiency of myoblast therapy in *mdx* mice [108]. Meanwhile, a lack of fibroblasts, a primary source of ECM components, also resulted in impaired muscle regeneration [109,110], suggesting that a fine balance of ECM components is required for maintenance of skeletal muscle. 

ECM contains collagen, laminin, or fibronectin, and is mostly produced by fibroblasts that exist in the interstitium of skeletal muscle. Since muscle satellite cells are anatomically close to interstitial fibroblasts, ECM is a principal factor controlling the activation and maintenance of myogenic cells. A previous study had demonstrated that collagen VI, an ECM protein, was critical as an extracellular niche protein to maintain the satellite cell pool [111]. Transplantation of fibroblasts from wild-type mice into *Col6a1-deficient* muscle could rescue resident muscle satellite cell self-renewal, indicating that ECM components, including collagen VI, can modulate satellite cell behaviour [111]. Similarly, Baghdadi et al. have recently identified that muscle satellite cells produce extracellular matrix collagen V in order to activate Notch signalling, which leads to maintenance of satellite cells in a quiescence state, in a cell-autonomous manner [112].

The basal lamina consists of extracellular matrix proteins, including laminins, fibronectin, and collagens. Laminins, which are a glycoprotein family consisting of α, β, and γ chains, are a critical component of the basal lamina [113]. Since muscle satellite cells are located between the plasma membrane and basal lamina in muscle fibres [16], a niche environment with laminins and integrin (α7β1), which are expressed in muscle satellite cells, potentially affects myogenic cell maintenance. Past studies reported that injection of laminin-111 (α1, β1, γ1) could improve dystrophic pathology in the muscle of a congenital myopathy model [114], and that exogenous laminin-111-treatment promotes proliferation and self-renewal in myogenic cells [115]. Similarly, Ishii et al., have found that reconstruction of the extracellular laminin environment using E8 domain of laminin (LM-E8), which was the minimally active fragment of laminin containing its integrin-binding sites, promotes proliferation in satellite cells by suppressing differentiation, and that recombinant LM-E8 treatment in myogenic cells from mice and humans could yield a high transplantation efficiency [116]. 

Fibronectin is also a vital component of the basal lamina and regulates muscle satellite cells. According to a previous study, Syndecan-4 and Fzd7 form a co-receptor complex in activated satellite cells, and binding of fibronectin to Syndecan-4 promotes Wnt signalling to promote the satellite stem cell expansion [117]. Additionally, knockdown of fibronectin manipulated by siRNA in isolated satellite cells severely impaired their ability to reconstitute the satellite cells, while in vivo overexpression of fibronectin could stimulate the expansion of satellite cells during muscle regeneration [117]. Given these observations, maintenance of the ECM environment in both donor cells and recipient muscle has the potential to enhance the efficacy of myogenic cell transplantation therapies greatly.

### 4.4. Rejuvenation

New strategies for obtaining sufficient numbers of transplantable cells aim to optimize in vitro conditions, which permit the expansion of myogenic cells that maintain an undifferentiated state. For example, one such strategy focuses on improving rejuvenation of aged myogenic cells through cell culture on substrates that mimic the in vivo muscle microenvironmental niche [118,119] and/or the utilization of small molecules to manipulate signalling pathways involved in myogenic cell proliferation [120,121,122,123]. Bernet et al. and Cosgrove et al. found that myogenic cells from aged mice had an impaired capacity to reconstitute myofibres and replenish stem cells in vivo following transplantation and that these impairments were due to elevated activity of the p38a and p38b mitogen-activated kinase (MAPK) pathway [121]. Furthermore, biochemical inhibition of p38a/b in aged myogenic cells could repair age-associated defects and improve the efficiency of cell transplantation with functional recovery [120,121]. Price et al. have demonstrated that the reduced capacity of muscle satellite cells in aged mice is related to an upregulation of Stat3 signalling [122], and that inhibition of Stat3 signalling by small molecules in aged cells in vitro enhances cell engraftment efficiency and self-renewal after transplantation [122]. Judson et al. recently demonstrated that methyltransferase Setd7 regulates the nuclear accumulation of β-catenin in proliferating myogenic cells and that genetic or pharmacological inhibition of Setd7 increases their expansion [123]. Furthermore, inhibition of Setd7 by the small molecule PFI-2 may enhance therapeutic potential after transplantation, including proliferation and self-renewal in mice and human myogenic cells [123].

Since MD patient-derived donor cells require genetic correction, these cells need to be expanded in vitro while maintaining myogenic cell potentials. Therefore, strategies to maintain or rejuvenate myogenic cells are necessary, and continued optimization of myogenic cell culture conditions using Notch-ligands, ECM treatment, or small molecules is needed.

## 5. Strategies to Improve or Maintain Myogenic Potential

In order to overcome the loss of myogenic cell function after expansion in vitro, as well as low transplantation efficiency in vivo, there is a demand for the development of coating materials that mimic the native cellular microenvironment. Recent studies have focused on simulating the niche-like environment using biomaterials and muscle resident cells, in order to accomplish better myogenic cell engraftment through the implantation of cells onto 3D scaffolds before transplantation [118,121]. Several types of biomaterials with different physical and chemical properties have been developed that enable tissue engineering for the treatment of muscle injury and MDs. Hydrogels are especially attractive biomaterials since they contain enough flexibility to adapt to changing environmental conditions, such as temperature, UV radiation, and pH fluctuations, and can be designed to mimic native tissue [124]. In addition, hydrogels have distinct advantages in muscles, including mild processing conditions, similarity to the native tissue microenvironment, and a non-invasive delivery method [125,126]. Therefore, an injectable hydrogel is an attractive tool to promote muscle regeneration or to deliver therapeutic agents including cells and/or bioactive molecules. For example, injection of hydrogels derived from decellularized muscle ECM that possesses a composition closely matched to native tissue, resulted in an increase in myogenic cell proliferation and capillary density [127], indicating that the maintenance of microenvironment using ECM components via hydrogel injection is potentially effective for use in cell therapies.

Rossi et al. demonstrated that the implantation of muscle satellite cells embedded within hyaluronan-based hydrogels into ablated mice muscle effectively restores the muscle tissue and enabled functional recovery [125]. Moreover, these reconstructed tissues contained a functional muscle satellite cell niche along with both neural and vascular networks [125], suggesting that combining hydrogel with myogenic cells could be a potential strategy for improving muscle environment and myogenic cell transplantation efficiency. A commonly used synthetic material, polyethyleneglycol (PEG), can easily be manipulated to modify its structure and mechanical properties for use in biomedical applications [124]. Gilbert et al. showed that muscle satellite cells cultured on soft PEG hydrogel substrate mimics the muscle elasticity of 12 kPa and promotes retention of self-renewal properties in vitro. In addition, myogenic cell transplantation efficiently contributed to muscle fibre reconstitution [118]. Cosgrove et al. demonstrated that aberrant p38α/β mitogen-activated protein kinase (MAPK) signalling in aged muscle satellite cells diminished self-renewal ability, while culture of aged muscle satellite cells on soft PEG hydrogels, along with chemical inhibition of p38 MAPK, led to a synergistic effect in stimulating expansion of engrafted myogenic cells [121]. Wang et al. showed that the delivery of myogenic cells along with growth factors VEGF and IGF-1, using a shape-memory, highly porous, biodegradable scaffold, promoted the regeneration of skeletal muscle tissue following severe muscle injury, resulting in functional recovery [128]. Furthermore, Fuoco et al. demonstrated that PEG-fibrinogen hydrogel-embedded mesoangioblasts overexpressing placental-derived growth factor, which enhances angiogenesis, improved survival and maturation of newly formed myofibres and generation of vessels and nerves [129]. Artificial muscles constructed with PEG-fibrinogen hydrogel-embedded mesoangioblasts resembled healthy muscles and led to the recovery of muscle morphology and functionality after implantation [129]. These studies indicate a potential for hydrogel use in cell therapies, although the issue of systemic delivery is still unresolved.

Construction of artificial human skeletal muscles through cell culture is a potentially beneficial tool for the study of pathological mechanisms of diseases, as well as for use in therapeutic transplants to replace impaired skeletal muscles [119,130] (Figure 1). Since artificial muscles contain vascular cells that mimic the microenvironment, they are able to considerably improve engraftment efficacy following implantation, via rapid anastomosis with host vessels [119,130]. Quarta et al. generated artificial muscle tissues by combining decellularized ECM scaffolds with freshly isolated muscle satellite cells and muscle resident cells containing fibro-adipogenic progenitors, macrophages, and endothelial cells. When these tissues were implanted into injured muscles, myofibre reconstitution and vascularization was observed, in addition to the recovery of force production [119]. Although muscles were incompletely innervated following bio-construct implantation, this condition was improved with exercise, resulting in the enhancement of force production [119]. Furthermore, Maffioletti et al. generated artificial skeletal muscle tissue on fibrin hydrogel using human pluripotent stem cells, including induced pluripotent stem cells (iPS cells) [130]. The constructed muscle tissues were implanted into muscles and were able to form muscle fibres with functional vessels [130]. A significant advantage of iPS cell use in the construction of artificial muscles is the feasibility of obtaining different types of cells from one cellular source, therefore enabling the generation of entire artificial muscle tissues containing all cellular constituents of skeletal muscles, including vascular endothelial cells, pericytes, and motor neurons, in addition to mimicking models of muscular disease with pathological hallmarks [130].

These developments represent significant advances towards validation of stem cell-based engineered tissues in the treatment of muscular diseases. Although problems remain, such as a need for larger tissues to replace entire DMD-affected muscles and the ability to reconstruct defective or locally affected smaller muscles, a reconstructive tissue engineering approach using myogenic cells has the potential for translation to clinical trials.

## 6. Concluding Remarks

Tremendous progress has been made in addressing the challenges of conventional cell therapy through a combination of gene modifications and bioengineering approaches. Such progress has contributed to overcoming some of the obstacles that have previously hindered the success of cell therapies. Nevertheless, significant limitations related to poor efficacy, unknown safety, and a lack of effective delivery of engrafted cells, still remain. Mesoangioblasts have great potential for use as a DMD therapy cell source since they can be systemically delivered and can contribute to muscle fibre construction. However, use of mesoangioblasts has not reached the clinic due to the low efficiency of cell transplantation [37]. Furthermore, in cases where cell therapy is combined with gene therapy, issues related to off-target effects or the immune response effect on genetically modified cells remain unsolved. 

Recent studies have provided evidence that cell transplantation therapy may not only produce DGC proteins but also may maintain or rejuvenate the microenvironment in skeletal muscle, limiting disease progression. Since the majority of fibres are lost and ageing muscle tissue is progressively replaced with connective and adipose tissue in dystrophic muscles, the general consensus is that most therapies would be ineffective at a late disease stage; therefore, any intervention should be conducted as early as possible. For example, exon-skipping is a promising approach to restore functional dystrophin and can be applied in the case of approximately 80% of all DMD mutations [131]. As we have previously reported, induction of exon-skipping through treatment of *mdx* mouse muscle with phosphorodiamidate morpholino oligomers (PMOs) while muscle regeneration was still active, led to efficient PMO delivery. This resulted in enhanced dystrophin expression, although this efficiency gradually declined with age [132]. As a result of ageing, the number of fibres was also significantly reduced. Therefore, maintenance or rejuvenation of the microenvironment in muscle stem cells and diseased muscles would be required to prevent disease progression and to enhance therapeutic effects. 

Recent studies have also revealed that changes in the muscle environment could be due to the impairment of muscle resident cells [133,134]. Lukjanenko et al. found that age-related failure of muscle regeneration was caused by the impairment of fibro-adipogenic progenitor (FAP) cells [133], which exist in the interstitial space of muscle and have the potential to differentiate into adipogenic and fibroblastic cells [135,136] (Figure 1). Systemic treatment with young FAP cells or with Wnt1-inducible signalling pathway protein 1, improved FAP-derived matricellular signal, leading to the restoration of myogenic cell capacity and muscle regeneration in aged mice [133]. Another study revealed that the appearance of adipocytes and muscle loss observed in limb-girdle muscular dystrophy 2B (LGMD2B) is caused by the accumulation of Annexin A2 (AnxA2) in FAP cells, and that pharmacological inhibition of adipogenesis can ameliorate muscle pathology caused by dysferlin deficiency [134]. These results suggest that restoring FAP function could be a therapeutic option to slow disease progression.

In addition to stem cell–based therapies, exosomes produced by stem cells can be employed as therapies. Several studies have suggested that cardiosphere-derived cells (CDCs) and CDC-secreted exosome vesicles (EVs) may show therapeutic potential in DMD [137,138,139,140,141]. CDCs, which are clonogenic and exhibit multilineage potential [137], have been shown to induce therapeutic regeneration of the infarcted human heart by shrinking scar size and promoting the growth of newly regenerated myocardium and vasculature, which are of endogenous origin [138]. Interestingly, transplantation of CDCs could be considered to support regeneration in the infarcted myocardium by indirect mechanisms, since long-term functional benefit and tissue regeneration persist long after all injected donor cells have been cleared immunologically [139]. Further studies have revealed that CDC-secreted EVs induce therapeutic regeneration, while the inhibition of EV production abolishes the benefit of CDCs [140]. Given these observations, CDCs and their EVs are potentially useful for the treatment of DMD cardiomyopathy and muscular degeneration. Aminzadeh et al. demonstrated that EVs secreted by CDCs, which were produced by cardiomyocytes of mice and in human DMD cardiomyocytes derived from iPS cells, can be systematically delivered to all tissues including heart, skeletal muscle, and brain [141]. Surprisingly, partial expression of full-length dystrophin in heart, diaphragm, and soleus of *mdx* mice was observed [141]. Furthermore, the suppression of liver inflammation, the functional benefits of cardiac injections of CDCs, and the continuous presence of EVs in the heart for at least three months in *mdx* mice [141] also suggest that CDCs could aid in delayed disease progression. However, further work is needed to clarify the role of CDC-derived EVs, since dystrophin protein restoration was transient, dystrophin protein or transcripts were absent, and it remains unclear whether the partial dystrophin restoration solely resulted from CDC-EVs [141]. 

In conclusion, an increased understanding of the microenvironment surrounding myogenic cells could lead to significant improvements in cell transplantation efficiency for MDs. We believe that cell transplantation therapy, combined with microenvironment improvement through the use of cell-based therapies, has excellent potential as a future therapeutic approach for MDs. If such a regenerative medicine approach is combined with precision gene therapy, this advanced treatment will likely be applicable to a wide range of both rare and common muscle disorders.

## Figures and Tables

**Figure 1 cells-08-01066-f001:**
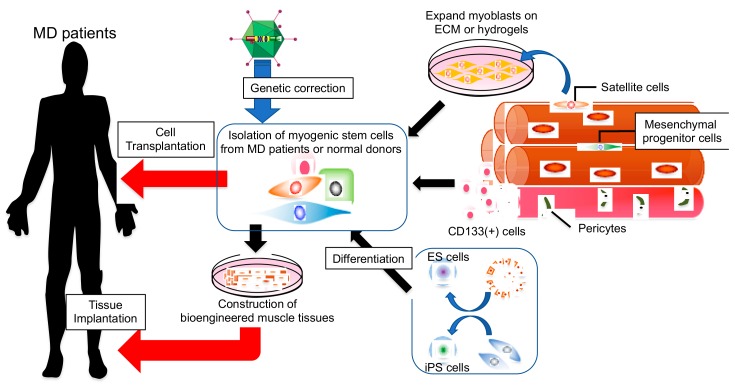
Therapeutic models for treatment of muscle diseases using myogenic stem cells. Muscle satellite cells, pluripotent stem cells (mesoangioblasts, and CD133+ cells) and ES/iPS cell-derived myogenic stem cells are possible cell sources for MD transplantation therapy. To generate a high yield of stem cells with myogenic potential, isolated cells are expanded by culture with ECM components or using hydrogel biomaterials. For allogeneic transplantation, isolated myogenic stem cells from MD patients are subjected to genetic correction. In addition to typical cell transplantation therapy, the implantation of bioengineered skeletal muscle tissues produced by myogenic stem cells is a promising approach.

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
