# Peer review of "Potential Therapies Using Myogenic Stem Cells Combined with Bio-Engineering Approaches for Treatment of Muscular Dystrophies"

_cells, 2019, doi:10.3390/cells8091066_

Round 1
Reviewer 1 Report
Line 41 'myofibrils retain a truncated form' I suggest to change to 'myofibrils retain an internally truncated form'.
I think that section 3 3. Cell sources for transplantation therapy, is not critical enough of the published work. You are listing all the cells that have been tried without indicating that very low (if any) success levels were obtained with some of these cells. I suggest to revise this part of the manuscrit.
Author Response
We are grateful to the reviewer for the comments and useful suggestions that have helped us to improve the manuscript. As indicated in the responses that follow, we have taken all these comments and suggestions into account in the revised version of our manuscript.
Line 41 'myofibrils retain a truncated form' I suggest to change to 'myofibrils retain an internally truncated form'.
(Response)
We added the term “an internally” in the text (see Page 2, Line 43).
I think that section 3 3. Cell sources for transplantation therapy, is not critical enough of the published work. You are listing all the cells that have been tried without indicating that very low (if any) success levels were obtained with some of these cells. I suggest to revise this part of the manuscript.
(Response)
Thank you very much for your suggestion. We carefully revised section 3 based on your suggestions (see revised section 3).

Reviewer 2 Report
In this review the Authors introduced a possibility of skeletal muscle regeneration based on combination of muscle stem/progenitor cells and gene editing methods as a potential therapy to treat muscular dystrophies.
However, before publication the Authors need to explain following points:
On the page 2 lines 88-93 the Authors stated that systemic delivery of satellite cells is not effective for cells migration trough the vessel endothelium and local intramuscular delivery will be beneficial for cell engraftment. However, at the end of this paragraph (page 3, lines 100-102) the Authors introducing positive results of treatment of DMD patients after systemic cells delivery of myogenic stem cells. Both information are true, however, should be better organized and explained eg. what kind of myogenic stem cells were used in cited studies [ref 33]. In the section 3 “Cell sources for transplantation therapy” the Authors described different muscle-derived stem/progenitor cells, however there is not clear if these cells are characteristic for mouse or human myogenic cells; according to the references it seems that most of introduced cells are of mouse-origin cells. This basic information on the stem/progenitor cells of muscle-origin is important for cell biology, and how it translated for human studies is important to present the current state of knowledge of stem cell therapy in human muscular dystrophies.Author Response
Response to Reviewer 2 Comments
We are grateful to the reviewer for the comments and useful suggestions that have helped us to improve the manuscript. As indicated in the responses described below, we have taken all these comments and suggestions into account in the revised version of our manuscript.
On the page 2 lines 88-93 the Authors stated that systemic delivery of satellite cells is not effective for cells migration trough the vessel endothelium and local intramuscular delivery will be beneficial for cell engraftment. However, at the end of this paragraph (page 3, lines 100-102) the Authors introducing positive results of treatment of DMD patients after systemic cells delivery of myogenic stem cells. Both information are true, however, should be better organized and explained eg. what kind of myogenic stem cells were used in cited studies [ref 33].
(Response)
Thank you for your suggestion. We have replaced “myogenic stem cells” by “mesoangioblasts”, and revised the sentences accordingly (see Page 3, Line 115).
In the section 3 “Cell sources for transplantation therapy” the Authors described different muscle-derived stem/progenitor cells, however there is not clear if these cells are characteristic for mouse or human myogenic cells; according to the references it seems that most of introduced cells are of mouse-origin cells. This basic information on the stem/progenitor cells of muscle-origin is important for cell biology, and how it translated for human studies is important to present the current state of knowledge of stem cell therapy in human muscular dystrophies.
(Response)
Thank you very much for your helpful comments. We fully agree with the reviewer to clarify the origin of transplanted cells and describe how the findings using the cells are applied to human studies. Based on the reviewer’s comments, we carefully modified the sentences in Section 3.

Reviewer 3 Report
Points to address
Title, the title does not adequately reflect the focus on the themes running through this review, namely bio-engineered approaches and role of cell-micro-environment interaction. So please make the title less generic. Page 1, line 4, there is an errant ‘1’ after the first author’s surname; Page 2, lines 45-48, the authors should add specific side effects beyond ‘adverse immune response’, particularly for steroid use e.g. bone and epidermal thinning, blood pressure, mood/behaviour changes, dysregulated metabolism, delayed puberty, stomach irritation etc. Page 3, lines 100-102, more details from the Cossu et al paper should be discussed e.g. what muscles were treated, adverse events etc. Page 3, section 3, does cell-survival/efficiency of cell-transplant vary for the different populations of myogenic cells? Also, what about discussing the issue of immune responses/ immunological tolerance to de novo dystrophin expression in revertant myofibres and CD4+/CD8+ T cell–mediated pathology (including points from the following references would be beneficial: Rosenburg et al. Sci Transl Med. 2015 Aug 5; 7(299): 299rv4; Flanigan et al. Hum Gene Ther. 2013 Sep; 24(9):797-806; Farini et al. Hum Gene Ther. 2013 Sep; 24(9):797-806). Page 3, line 139, Does the ability of CD133+ cells and mesoangioblasts to penetrate vessel walls have any tumourigenicity concerns or inappropriate migration to unwanted tissue sites? Page 4, line 155, human limb-girdle muscular dystrophy type 2D and other muscular dystrophy types e.g. myotonic, Emery-Dreifuss, Fukuyama congenital muscular dystrophy should be better introduced in section 1. Also could the authors add a comment on whether myogenic cell therapy would be a generic approach suitable for all types of MD or whether one cell therapy product would better suit/or not a particular MD type. Page 4, line 167, please add 3 lines and appropriate references on the following points related to this topic: iPS cells should not be present, heterogeneity of cells derived from iPS cells e.g. functional immaturity, methods used for clinical-grade therapies to detect un-wanted cells/cell functions. Page 6, line 245-293, in section 4.3. on ECM the authors should discuss the pros and cons of current clinical/pre-clinical pharmaceutical agents targeting ECM/fibrosis e.g. phosphodiesterase inhibitors (NCT01359670 and Nelson et al. Neurology. 2014 Jun 10;82(23):2085-91. doi:10.1212/WNL.0000000000000498), blocking cytokine signalling (De Paepe et al. Muscle Nerve. 2012 Dec;46(6):917-25. doi: 10.1002/mus.23481), histone deacetylase inhibition (Milan et al. Cell Death and Disease (2018) 9:108 DOI 10.1038/s41419-017-0174-5), immuno-proteasome inhibition (Farini and Gowran et al Am J Pathol. 2019 Feb;189(2):339-353. doi:10.1016/j.ajpath.2018.10.010) etc. Section 4 (all sections), many of the factors that impact myogenic potential already have targeting drugs that have in many cases reached advanced clinical trial stage. Therefore, this section as a whole could benefit from extra information linking the clinical trials/potential of each therapeutic target discussed in each sub-section and why cell-therapy is better or how the drug can be combined with cell therapy to enhance the therapeutic effect. Section 6: since more new studies are discussed in this section, calling it ‘Concluding Remarks’ would be a better fit than ‘summary and perspective’.
General edits
Page 2, line 61, please add allogeneic before ‘myoblasts’; Page 2, line 69-71, were the italics intended?; Page 2/3, line 60/118, mdx or Mdx; Page 3, line 129, insert ‘were’ between ‘brother’ and ‘transplanted’’; Page 3, line 132, change to: although; Page 3, line 134, Scid should be SCID; Page 3, line 136, UK English is used in the majority of the manuscript so please check for continuity e.g. myofibres not myofibers, ageing not aging etc.; Page 4, line 152, remove commas and place a space between ‘generatingan’ from ‘the low risk…..rejection, and the potential for generatingan unlimited number of cells,’; Page 4, line 172, use a small ‘m’; Page 5, line 182, change to: decrease; Page 5, line 183, change to: capacity; Page 5, line 189, change to: myogenic cell function declines with age; Page 5, line 190-191, change to: which shorten telomeres and impair self-renewal mechanisms; Page 5, line 194, change to: loss of myogenic cell function; Page 5, line 195, change to: mouse models of ageing; Page 5, line 196, replace with: to; Page 5, line 213, change to: that instead leads to; Page 5, line 222, change to: although; Page 8, line 357, change to: although; Page 8, line 361, change to: contain; Page 9, line 420, change to: as; Page 9, line 424/427, why the underlined text?; Page 10, line 435, add Furthermore before ‘the suppression of liver…’.

Author Response
Response to Reviewer 3 Comments
We are grateful to the reviewer for the useful suggestions that have helped us to improve the manuscript. As indicated in the responses described below, we have taken all the comments and suggestions into account in the revised version of our manuscript.
Points to address
Title, the title does not adequately reflect the focus on the themes running through this review, namely bio-engineered approaches and role of cell-micro-environment interaction. So please make the title less generic.
(Response)
Thank you for your suggestion. As the reviewer mentioned, the previous title “The Potential of Myogenic Stem Cells for Treatment of Muscular Dystrophies” has been replaced by “Potential Therapies using Myogenic Stem Cells combined with Bio-engineered Approaches for Treatment of Muscular Dystrophies”.
Page 1, line 4, there is an errant ‘1’ after the first author’s surname;
(Response)
We deleted “1” after the first author’s surname.
Page 2, lines 45-48, the authors should add specific side effects beyond ‘adverse immune response’, particularly for steroid use e.g. bone and epidermal thinning, blood pressure, mood/behavior changes, dysregulated metabolism, delayed puberty, stomach irritation etc.
(Response)
Thank you very much for your suggestion. We added much information to discuss the specific side effects of steroids (see Page 2, Line 55-58).
Page 3, lines 100-102, more details from the Cossu et al paper should be discussed e.g. what muscles were treated, adverse events etc.
(Response)
Thank you very much for your comment. We extensively discussed the paper written by Cossu et al. accordingly (see Page 3, Line 114-117 and Page 4, Line 181-186)
Page 3, section 3, does cell-survival/efficiency of cell-transplant vary for the different populations of myogenic cells? Also, what about discussing the issue of immune responses/ immunological tolerance to de novo dystrophin expression in revertant myofibres and CD4+/CD8+ T cell–mediated pathology (including points from the following references would be beneficial: Rosenburg et al. Sci Transl Med. 2015 Aug 5; 7(299): 299rv4; Flanigan et al. Hum Gene Ther. 2013 Sep; 24(9):797-806; Farini et al. Hum Gene Ther. 2013 Sep; 24(9):797-806).
(Response)
Thank you very much for your suggestions. As the reviewer suggested, since the infiltration of dystrophin reactive T cells was observed in DMD muscles, the presence of T cell immunity may affect the cell survival or the efficacy of cell transplantation. According to previous studies, myoblasts are considered to have the capacity to trigger T-cell mediated immune response in regenerating muscle due to the increase of MHC (major histocompatibility complex) I/II expression induced by inflammatory factor including IFNg (Mantegazza et al., Neurology 1991; Englund et al., Am J Pathol 2001; Rosenberg et al., Sci Transl Med 2015). On the other hand, mesoangioblasts and iPS cell-derived mesoangioblast-like cells have shown the induction of PD-1/PD-L1, the suppressor of immune response, and peculiar resistance to T-cell killing, by the stimulation of proinflammatory cytokines (English et al., Stem Cells Dev 2013; Li et al., F1000Research 2013). These observations would affect the differences of the cell survival following cell transplantation among the myogenic cell. We add the sentences to discuss the relevance between the cell-survival/cell-transplantation efficiency and immune responses (See page 4, Line160-171).
Page 3, line 139, Does the ability of CD133+ cells and mesoangioblasts to penetrate vessel walls have any tumourigenicity concerns or inappropriate migration to unwanted tissue sites?
(Response)
According to previous studies, any adverse events including inflammation, tumour formation or tissue infarction were not reported after the intra-arterial injection of human mesoangioblast (Cossue, 2015) and the intramuscular injection of human muscle CD133+ cells (Torrente, 2007). However, possible concern is that the delivered cells are trapped in the filter organs. According to previous studies, more than 30% of mesoangioblasts were detected in the liver, lungs or spleen of a-sarcoglycan deficient mice 24 hours after arterial injection, and their numbers were still high even in 8 months after injection (Galvez et al., 2006). Further, about 90% of mesoangioblasts or pericytes from human muscles were localized in filter organs of SCID-mdx mice 24 hours after arterial injection (Dellavalle et al., 2007). In related to these problems, the trapped mesenchymal stem cells after abdominal aorta injection caused the interruption of blood flow through the microvasculature downstream of the site of injection, resulted in pulmonary embolism and subsequent death in up to 40% of mice (Furlani et al., 2009). These observations indicate that arterial delivery of myogenic stem cells also may disrupt arterial or microvascular blood flow, leading to tissue damage. We included these concerns into section 3 (See Page 4, Line 172-180).
Page 4, line 155, human limb-girdle muscular dystrophy type 2D and other muscular dystrophy types e.g. myotonic, Emery-Dreifuss, Fukuyama congenital muscular dystrophy should be better introduced in section 1.
(Response)
Thank you very much for your suggestion. We have added the sentences to explain other muscular dystrophy types in section 1 (see revised Section 1).
Also could the authors add a comment on whether myogenic cell therapy would be a generic approach suitable for all types of MD or whether one cell therapy product would better suit/or not a particular MD type.
(Response)
Thank you very much for your comments. Although myogenic cell therapy is still challenging for several types of muscular dystrophy, in which the majority of muscles are affected, muscular dystrophies with distal muscle involvement including oculopharyngeal muscular dystrophy, may benefit from a local injection of myogenic cells. We modified the sentences to emphasize this point (See Page 3, Line 104-108).
Page 4, line 167, please add 3 lines and appropriate references on the following points related to this topic: iPS cells should not be present, heterogeneity of cells derived from iPS cells e.g. functional immaturity, methods used for clinical-grade therapies to detect un-wanted cells/cell functions.
(Response)
We added the sentences to explain the possible concerns of iPS-derived myogenic cell transplantation (See Page 5, Line 211-216).
Page 6, line 245-293, in section 4.3. on ECM the authors should discuss the pros and cons of current clinical/pre-clinical pharmaceutical agents targeting ECM/fibrosis e.g. phosphodiesterase inhibitors (NCT01359670 and Nelson et al. Neurology. 2014 Jun 10;82(23):2085-91. doi:10.1212/WNL.0000000000000498), blocking cytokine signalling (De Paepe et al. Muscle Nerve. 2012 Dec;46(6):917-25. doi: 10.1002/mus.23481), histone deacetylase inhibition (Milan et al. Cell Death and Disease (2018) 9:108 DOI 10.1038/s41419-017-0174-5), immuno-proteasome inhibition (Farini and Gowran et al Am J Pathol. 2019 Feb;189(2):339-353. doi:10.1016/j.ajpath.2018.10.010) etc.
(Response)
Thank you for your suggestions. We added the sentences and references to discuss ECM/fibroblasts as the clinical targets (See Page 7, Line 333-345).
Section 4 (all sections), many of the factors that impact myogenic potential already have targeting drugs that have in many cases reached advanced clinical trial stage. Therefore, this section as a whole could benefit from extra information linking the clinical trials/potential of each therapeutic target discussed in each sub-section and why cell-therapy is better or how the drug can be combined with cell therapy to enhance the therapeutic effect.
(Response)
Thank you very much for your suggestion. We have inserted the additional sentences and references in order to discuss the combined therapies using myogenic stem cells and clinically available drugs (See new Section 3 and 4).
Section 6: since more new studies are discussed in this section, calling it ‘Concluding Remarks’ would be a better fit than ‘summary and perspective’.
(Response)
We have replaced “summary and perspective” by Concluding Remarks” (See page 10, Line 467).
General edits
Page 2, line 61, please add allogeneic before ‘myoblasts’
(Response)
We added the term “allogeneic” in the text (See page 2, Line 73).
Page 2, line 69-71, were the italics intended?
(Response)
No. These sentences were revised (See page 2, Line 80-83).
Page 2/3, line 60/118, mdx or Mdx
(Response)
Mdx was written as mdx in the whole mauscript.
Page 3, line 129, insert ‘were’ between ‘brother’ and ‘transplanted’’
(Response)
We added the term “were” in the text (See page 4, Line 149)
Page 3, line 132, change to: although
(Response)
We changed “though” to “although” (See page 4, Line 154)
Page 3, line 134, Scid should be SCID
(Response)
We replaced “Scid” by “SCID” (See page 4, Line 156 and 158).
Page 3, line 136, UK English is used in the majority of the manuscript so please check for continuity e.g. myofibres not myofibers, ageing not aging etc.
(Response)
We replaced “myofiber” by “myofibre”, and “aging” by “ageing”.
Page 4, line 152, remove commas and place a space between ‘generatingan’ from ‘the low risk…..rejection, and the potential for generatingan unlimited number of cells,’
(Response)
We deleted commas and revised the sentence accordingly (See page 4, Line 196)
Page 4, line 172, use a small ‘m’
(Response)
We used small “m” instead of large “M”(See page 5, Line 236).
Page 5, line 182, change to: decrease;
(Response)
We changed “decreases” to “decrease” (See page 6, Line 247)
Page 5, line 183, change to: capacity
(Response)
We replaced “capacities” with “capacity” (See page 6, Line 248).
Page 5, line 189, change to: myogenic cell function declines with age
(Response)
We revised the sentences as the reviewer pointed out (See page 6, Line 254)
Page 5, line 190-191, change to: which shorten telomeres and impair self-renewal mechanisms
(Response)
We revised the sentences as the reviewer pointed out (See page 6, Line 255).
Page 5, line 194, change to: loss of myogenic cell function
(Response)
We replaced “myogenic cell loss of function” by “loss of myogenic cell function” (See page 6, Line 259).
Page 5, line 195, change to: mouse models of ageing
(Response)
We replaced “aging mouse models” by “mouse models of ageing” (See page 6, Line 260)
Page 5, line 196, replace with: to
(Response)
We replaced “in” by “to” (See page 6, Line 261)
Page 5, line 213, change to: that instead leads to
(Response)
We replaced “leading to” by “that instead leads to” (See page 6, Line 283-284)
Page 5, line 222, change to: although
(Response)
We replaced “though” by “although” (See page 6, Line 292).
Page 8, line 357, change to: although
(Response)
We replaced “though” by “although” (See page 9, Line 441)
Page 8, line 361, change to: contain
(Response)
We replaced “contains” by “contain” (See page 9, Line 445)
Page 9, line 420, change to: as
(Response)
We replaced “in” by “as” (See page 11, Line 504)
Page 9, line 424/427, why the underlined text?
(Response)
Thank you very much for your suggestion. These underlines were removed (See page 11, Line 508/511).
Page 10, line 435, add Furthermore before ‘the suppression of liver…’.
(Response)
We added the term “Furthermore” (See page 11, Line 519).

Round 2
Reviewer 3 Report
The manuscript is suitable for publication subject to basic copy editing etc., by the publishers.